# Relevance of Medullary Vein Sign in Neurosarcoidosis

Richard Liberio [1,2], Emily Kramer [1,2], Anza B. Memon [3], Ryan Reinbeau [1,2], Parissa Feizi [4], Joe Joseph [4], Janet Wu [2] and Shitiz Sriwastava [1,2,4,5,6,*]

[1] School of Medicine, West Virginia University, Morgantown, WV 26506, USA
[2] Department of Neurology, West Virginia University, Morgantown, WV 26506, USA
[3] Department of Neurology, Henry Ford Hospital, Detroit, MI 48202, USA
[4] Department of Neuroradiology, West Virginia University, Morgantown, WV 26506, USA
[5] Department of Neurology, Wayne State University, Detroit, MI 48201, USA
[6] West Virginia Clinical and Translational Science Institute, Morgantown, WV 26506, USA
* Correspondence: shitiz.sriwastava@hsc.wvu.edu; Tel.: +1-304-581-1903

**Abstract:** Background: Central nervous system involvement is uncommon in patients with sarcoidosis. It remains a diagnostic challenge for clinicians, as there is a broad differential diagnosis that matches the presenting neurological signs. Often, the imaging findings also overlap with other disease entities. One understudied finding in patients with neurosarcoidosis is the presence of medullary vein engorgement on SWI imaging, termed the "medullary vein sign", which has been postulated to be a specific sign for neurosarcoidosis. This study aims to provide an understanding of the diagnostic potential of the medullary vein sign. Methods: Thirty-two patients who presented with neurologic signs concerning for possible neurosarcoidosis were analyzed retrospectively for the presence of the medullary vein sign. Results: Out of these cases, 7 cases of definitive neurosarcoidosis cases were found based on other imaging signs, biopsy and CSF analysis; the remaining were classified into groups as possible (16), probable (5) and (4) cases of other infectious meningoencephalitis including 2 cases of autoimmune encephalitis. Seven patients among all of these cases were found to have the medullary vein sign on imaging, with five cases with confirmed and two cases from possible neurosarcoidosis. The sensitivity of the medullary vein sign in this study was 71.4%, and the specificity was 92.3%. Discussion: The benefits of improving diagnostic criteria for neurosarcoidosis include more rapid diagnosis leading to more prompt treatment, less exposure to potentially harmful antibiotics or antifungals, and less long-term neurological effects. Our results support that the medullary vein sign will potentially fill in the diagnostic gaps that have challenged the timely diagnosis of neurosarcoidosis. Conclusions: Our findings support that the medullary vein sign has a high specificity and should be included in the diagnostic criteria for neurosarcoidosis.

**Keywords:** MRI; neurosarcoidosis; medullary vein sign; neuroimaging

## 1. Introduction

Sarcoidosis is a relatively rare disease that affects more than 150,000 Americans. It is characterized by a cell-mediated granulomatous response to unidentified antigens, which results in the disruption of normal tissue microarchitecture [1,2]. Diagnosis is achieved via biopsy of affected tissue, showing noncaseating granulomas, usually lung or skin. Granulomatous inflammation in sarcoidosis can affect any organ system, most commonly the lungs, but also the nervous system [1,3].

Neurosarcoidosis affects 3–10% of individuals with sarcoidosis, and 75% of this subset will develop neurological symptoms within the first two years of being diagnosed with sarcoidosis. However, up to 80% of patients with neurosarcoidosis will present with neurological symptoms prior to the diagnosis of sarcoidosis [4–6]. Neurosarcoidosis can affect any part of the central nervous system; however, the most common presentation is

cranial neuropathy. Other neurological structures involved include the meninges, brain parenchyma, neuroendocrine glands, spinal cord, and peripheral nerves [6,7].

Diagnosis is a pervasive clinical challenge in the management of neurosarcoidosis. Diagnosing neurosarcoidosis relies on clinical signs, imaging, and CSF analysis. Historically, the diagnosis is made clinically with proposed diagnostic criteria that varies between physicians and institutions. Diagnostic criteria variability thus leads to a lack of specificity and sensitivity depending on physician or institutional preferences. Current proposed diagnostic criteria place a patient in three risk categories for neurosarcoidosis: possible, probable, and definite. However, variability exists between diagnosticians in each of these categories [6,7].

The imaging of choice for neurosarcoidosis work up is MRI with gadolinium, with pathology of neurosarcoidosis affecting the brain parenchyma, leptomeninges, dura mater, or spinal cord. The most common imaging finding in neurosarcoidosis is multiple non-enhancing periventricular white matter lesions. The most specific is the "trident sign," in the dorsal cord region, which is thought to relate to perivascular spread of leptomeningeal granulomatous inflammation [5,8,9]. Cerebrospinal fluid (CSF) analysis is also non-specific, although elevated levels of angiotensin converting enzyme (ACE) in the CSF is common in neurosarcoidosis. However, the false-positive and false-negative rate is high. Only 33–50% of patients with elevated CSF ACE have neurosarcoidosis according to the literature [8,10]. The differential based on the above diagnostic findings is thus quite broad and includes Neurosarcoidosis, CNS lymphoma, multiple sclerosis, infectious disease, autoimmune disease, osmotic demyelination, and other demyelinating diseases [10].

Treating neurosarcoidosis involves first-line intravenous corticosteroids in an effort to reduce granulomatous inflammation within the nervous system, just as sarcoidosis would be treated outside the nervous system. Second-line treatments include methotrexate, azathioprine, mycophenolate mofetil, and chloroquine [3,10]. A favorable outcome was reported in 71% of patients receiving corticosteroid monotherapy. In a meta-analysis performed by Fritz et al., 27% of patients reached remission, 32% achieved incomplete remission, 24% had stable pathology, and 6% deteriorated. The delay in treatment initiation may have significant implications on disease progression and chances of remission [11]. Further updates on treatment guidelines were recently revised by the European Respiratory Society in 2021 [12].

The literature for neuroimaging findings specific to neurosarcoidosis is relatively scarce. An association between existing neurosarcoidosis and medullary vein engorgement has only been cited once, but the imaging finding itself is uncommon to other pathology outside of medullary vein thrombosis, dural AV fistula, and rarely systemic lupus erythematosus [9,11]. The only case series on this topic cites 33% (7 of 21) of patients with neurosarcoidosis to demonstrate medullary vein engorgement on susceptibility weighted magnetic resonance imaging (SWI) [7]. In patients with isolated neurologic symptoms, this imaging finding can be important in contributing to the more efficient diagnosis of neurosarcoidosis in the presence of supporting signs. This study aims to provide the relevance of potential of the medullary vein sign in neuroimaging marker in neurosarcoidosis.

## 2. Materials and Methods

A single center retrospective study of patients recruited at the Department of Neurology, West Virginia University (WVU) was performed between 1 August 2022 and 31 March 2022. We have included 7 definitive neurosarcoidosis cases in this study from amongst a total of 32 patients based on inclusion criteria of confirmed diagnosis of neurosarcoidosis. Among them, 7 of these cases had the characteristic medullary vein sign on SWI/GRE sequences on MRI brain imaging (Figure 1). Data analysis included age, sex, symptoms, risk factors, serology status for antibodies, imaging findings, treatment, and recovery (Table 1). Two neuroradiologists independently performed the neuroimaging study. Neuroimaging findings and pathophysiology comparisons that were considered within the scope of our study were mainly focused on medullary vein signs and leptomeningeal and

pachymeningeal enhancement (Figures 1 and S1). In this study, MRI images were reviewed using SWI/GRE sequences in a retrospective search for medullary vein engorgement, termed medullary vein sign. Of the total 32 patients reviewed, 7 were confirmed to have neurosarcoidosis through biopsy and diagnosis of exclusion. Sensitivity was calculated as the number of correctly identified cases of neurosarcoidosis based on medullary vein sign divided by the total number of patients in this study that were found to have confirmed neurosarcoidosis. Specificity was calculated as the number of patients in the study who did not have neurosarcoidosis or the medullary vein sign divided by all individuals without neurosarcoidosis.

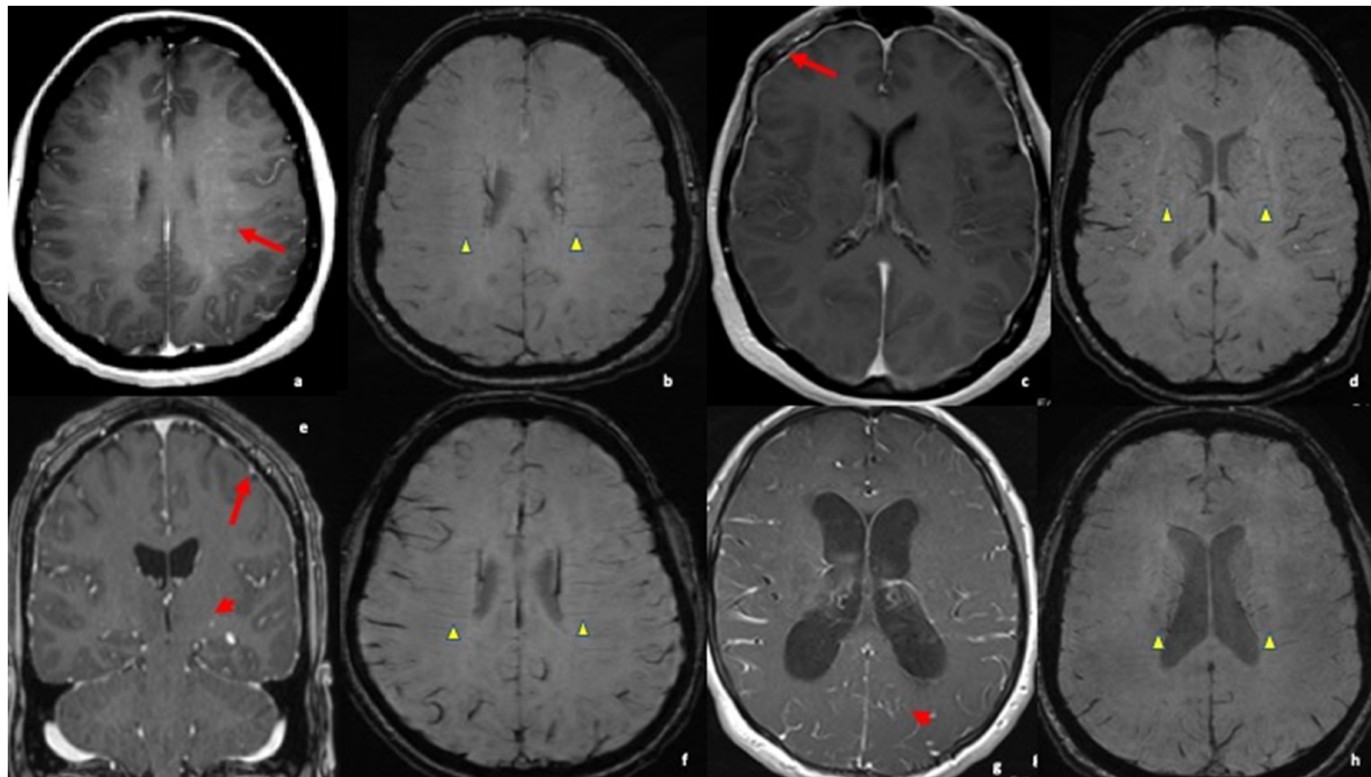

**Figure 1.** MRI axial (**a**) of the gadolinium-enhanced brain demonstrating perivascular enhancement throughout (red arrow); MRI brain (**b**,**d**,**f**,**h**) suspected weight images (SWI) show mildly dilated medullary veins perpendicular to long axis of lateral ventricles (yellow arrowhead); (**c**) gadolinium-enhanced MRI of the brain shows diffuse pachymeningeal enhancement (red arrow); (**e**) coronal and (**g**) axial cuts in post-gadolinium-enhanced MRI of the brain show leptomeningeal enhancement (red arrowhead) and pachymenigeal enhancement (red arrow).

**Table 1.** MRI Imaging and image analysis findings. The table lists five cases of confirmed neurosarcoidosis with a positive medullary vein sign upon review of MRI imaging along with clinical presentation, MRI findings, CSF analysis, treatment, and clinical outcome.

| Case | Age/Gender | Year of Diagnosis | Onset/Presentation | CSF Analysis | MRI Results | Treatment | Outcomes |
|------|------------|-------------------|--------------------|--------------|-------------|-----------|----------|
| 1 * | 56 y.o Female | 2015 | Confusion, thought blocking, leg weakness | WBC: 50 70% lymphocytes; 30% mononuclear Protein: 57 mg/dL | Small focus of restricted diffusion along left lateral medulla Medullary vein sign + | Outpatient follow-up Risk factor reduction | Resolution of acute confusion Residual facial numbness and leg weakness No progression |
| 2 * | 34 y.o. female | 2013 | Intractable headache Weakness Nausea/Vomiting Vision change | CSF ACE: normal Serum ACE: 137 U/L Glucose: 44 mg/dL WBC: 10 Lymphocytes: 79% PMNs: 10% Monos: 7% | Nonspecific markedly abnormal appearance of nodular enhancement and thickening along the pial surface in posterior fossa and basal cisterns and along tentorium and interhemispheric fissure. Developing hydrocephalus Medullary Vein sign + | IV solumedrol 1 g for 3 days Followed by 40 mg prednisone PO Methotrexate 10 mg weekly VP Shunt | No show to several appts. Headaches/vomiting unresolved but vision problems improved |
| 3 * | 62 y.o. Female | 2015 | Double vision, facial paresthesia, proprioception/balance deficits | CSF ACE: normal CSF Glucose: 59 mg/dL Protein: 41 mg/dL WBC: 5 | Diffuses dural thickening with marked dural enhancement in parasellar region Medullary vein sign + Mild leptomeningeal enhancement of optic nerves | 1000 solumedrol for four days, followed by PO steroid 60 mg for one month Methotrexate Rituximab Q6 months | Continued headaches after cessation of steroids. Visual problems after cessation of steroids. Was restarted on a steroid taper. Subsequently weaned off steroids and started on Methotrexate in 2017, which improved headaches and visual symptoms |
| 4 * | 32 y.o. female | 2018 | Intermittent monocular blindness and blurry vision Headache Vertigo N/V Numbness | CSF protein: 66 mg/dL CSF Glucose: 54 mg/dL WBC: 6 35% neutrophil 48% lymphs | Scattered nonenhancing parenchymal lesions in white matter of cerebral hemispheres Medullary vein sign + | 1 g Solumedrol IV TI, transitioned to PO prednisone taper 1 dose IV methotrexate transitioned to 25 mg oral once per 1 week | Improvement in N/V, headaches, and vertigo. Still some residual vision impairment treated with subcutaneous methotrexate per her ophthalmologist |

**Table 1.** *Cont.*

| Case | Age/Gender | Year of Diagnosis | Onset/Presentation | CSF Analysis | MRI Results | Treatment | Outcomes |
|------|-----------|-------------------|---------------------|--------------|-------------|-----------|----------|
| 5 * | 33 y.o. Female | 2018 | Papilledema Neck pain New blurry vision headaches | Protein: 216 mg/dL Glucose: 38 mg/dL WBC: 7 neutrophil % 1 Lymphocyte % 70 Monocytes % 27 ACE CSF: 3.5 | Scattered bilateral cerebral white matter lesions, some with associated enhancement. Leptomeningeal enhancement Medullary Vein sign + | Prednisone taper transition to imuran | Complete resolution of symptoms |
| 6 ** | 27/y o Female | 2019 | Left upper and lower extremity weakness | None | Multiple ring-enhancing lesions on post contrast T1 images, the largest within the right basal ganglia and the pons | 1 g Solumedrol IV TI, transitioned to PO prednisone taper | stable |
| 7 ** | 48/yo Male | 2018 | of bilateral retro-orbital pain and progressive vision loss over the course of two months | CSF ACE mildly elevated at 3.7 U/L (normal up to 2.5), 12 WBC predominant lymphocytic 90%, and negative meningitis panel | Enhancing left cerebellar nodule with surrounding abnormal hyperintense signal on FLAIR extending to the cerebellar peduncle. Mass-like enhancement of the folia surrounding the nodule was also noted. MRI orbits showed bilateral smooth optic nerve sheath enhancement without abnormal signal within the optic nerves | on 1 dose of cyclophosphamide with Mesna and discharged with oral tapering prednisone | Partial improvement in vision |

* Definitive neurosarcoidosis; ** Possible neurosarcoidosis; ACE, Angiotensin-converting enzymes; PMN, Polymorphonuclear neutrophil.

## 3. Results

### 3.1. General Results

Among the 32 patient charts reviewed at WVU, a total of 7 were classified as definite neurosarcoidosis based on tissue biopsy with (including three cases of cervical spine biopsy proven sarcoidosis) additional diagnostic labs including CSF and MRI imaging, 16 were possible neurosarcoidosis based on CSF and MRI findings, and 5 were probable neurosarcoidosis based on extra neuronal tissue biopsy (hilar lymphadenopathy biopsy with lung sarcoidosis, skin and bone biopsy). Among the 32 patients, 4 were found to have a different etiology for neurological symptoms other than neurosarcoidosis (1 bacterial menigoencephalitis;1 fungal encephalitis; 2 CNS autoimmune encephalitis). In 28 out of 32 possible/definitive neurosarcoidosis cases, 17 were female, and 11 were male. The most common symptoms involved in the patients with definite/possible neurosarcoidosis were progressive vision changes, limb weakness, and headache. The average age of presenting symptoms was 55 years old, spanning from 27 to 83 years old. The most common treatments were intravenous methylprednisone followed by prednisone taper (25%), methotrexate and rituximab (12%), with few patients receiving azathioprine (1%); treatment information for the rest of the cases was not available. There were no fatal cases reported, and patients were stable following treatment of the presenting symptoms. Ten patients presenting with symptoms of neurosarcoidosis had a prior diagnosis of tissue biopsy of either lung sarcoidosis or non-caseating granuloma of skin or bone.

The sensitivity of the medullary vein sign in this study was 71.4% and the specificity was 92.3% among patients with either a definite diagnosis or definite rule out of neurosarcoidosis (total of 32 patients). The sensitivity of the medullary vein sign in all reviewed patients including those with suspected but not confirmed (counting as a positive case of neurosarcoidosis), and definite neurosarcoidosis, and the number of patients who were found not to have neurosarcoidosis was lower at 25% (total of 28/32 patients).

### 3.2. CSF Findings

Patients with possible, probable or definite neurosarcoidosis had mostly inconclusive CSF findings except for three patients who had elevated CSF ACE levels. Total cell counts were mildly elevated in 13 of 32 cases with predominant lymphocytosis. Ten cases had elevated protein levels of >50 mg/dL. Patients without diagnosis of neurosarcoidosis who had neurologic symptoms due to another etiology had elevated cell counts in CSF including lymphocytes >80% or neutrophil counts greater than 75% (four cases).

### 3.3. MRI Findings

MRI findings in patients with possible or definite neurosarcoidosis were mostly leptomeningeal enhancement, cord lesions, or scattered parenchymal lesions. On chart review, MRI findings were not a deciding factor in diagnosis in many cases. MRI findings consistent with possible neurosarcoidosis led to brain biopsy in only three cases, all of which confirmed neurosarcoidosis. Seven total cases had medullary vein engorgement upon retrospective image analysis as part of this study.

## 4. Discussion

Sarcoidosis has been shown to manifest itself in the nervous system in about 3–10% of patients with systemic sarcoidosis [1]. Difficulties in the diagnosis are largely attributed to debates in proper diagnostic criteria. The presenting symptoms of neurosarcoidosis are similar to several other pathologies such as fungal, bacterial or viral meningitis, other autoimmune diseases, or demyelinating diseases. These issues raise a concern for delayed diagnosis and treatment initiation and necessitate other means of differentiating between neurosarcoidosis and other pathologies, especially in individuals who have not had a prior diagnosis of sarcoidosis in other organ systems [13–19].

Susceptibility weighted imaging (SWI) is a largely accessible neuroimaging tool that has shown characteristic results in patients with neurosarcoidosis. Deep medullary vein

engorgement has been associated with neurosarcoidosis as well as other autoimmune diseases such as systemic lupus erythematosus [14,20–23]. This finding is hypothesized to be caused by a chronic inflammatory reaction. Over time, this immune response causes microvascular damage, which can be seen as localized white matter injury and vessel dilation [21]. To date, there has not been sufficient research on the utility of SWI imaging related to diagnosing diseases such as neurosarcoidosis.

SWI imaging has also been recognized as a sensitive and specific test for brain venous malformations [20,24]. This pathology is visualized on SWI as hyperintensities [25]. However, the clinical picture is much different in neurosarcoidosis and brain venous malformation. Venous malformations present with headache and continue to progress as bleeding continues [20].

Deep medullary veins are easily visualized on SWI as hypointense linear structures perpendicular to the lateral ventricles. As seen in this study, as well as other studies, medullary vein engorgement will present as enlarged, tortuous veins that are usually symmetric and near the level of the corona radiata [19,21]. A previous study by Zamora et al. [19] also demonstrated the finding of deep medullary vein engorgement in patients with neurosarcoidosis and further found that these patients were more likely to have perivascular space involvement. This finding may support the pathogenesis of sarcoidosis spread to the CNS, which is thought to occur through hematogenous spread with eventual dissemination into the perivascular spaces [16].

Though the sensitivity is relatively low, the medullary vein sign should be used as an adjunct to other previously described criteria for neurosarcoidosis as mentioned above. Rather than being used as a diagnostic criterion alone, it is best to use this finding as an adjunct in supporting the diagnosis of neurosarcoidosis. It should be used to inform radiologists to actively look for this sign when neurosarcoidosis is being considered in the differential diagnosis. It is worth reiterating that none of the patients who were retrospectively found to have signs supportive of medullary vein engorgement had any mention of this findings in the initial MRI dictation. This supports the idea that revised guidelines for the diagnosis and treatment of neurosarcoidosis should include a review of SWI/GRE sequences to assess for the presence of the medullary vein sign. The presence of the medullary vein sign should be added to the previously studied MRI findings in neurosarcoidosis such as leptomeningeal enhancement, hydrocephalus, and parenchymal lesions that are sensitive but not specific to neurosarcoidosis [15,22,23]. In our study, the specificity of medullary vein engorgement is 92.3% filling in a diagnostic gap in the investigation of possible neurosarcoidosis. Therefore, in patients with suspected neurosarcoidosis, SWI imaging, which is more sensitive than GRE, should be utilized in the search for a medullary vein sign.

As mentioned previously, the differential diagnosis in patients with neurological signs concerning for neurosarcoidosis includes fungal, viral, and bacterial infections [17,18]. Six of the ten patients with confirmed neurosarcoidosis were initially treated with antibiotics or antifungals due to erroneous diagnosis of infection. These patients retrospectively had confirmed medullary vein signs on imaging. In these cases, SWI imaging demonstrating the medullary vein sign would have potentially prevented the unnecessary use of antibiotics and antifungals. The earlier diagnosis of neurosarcoidosis would also expedite the time to initiation of systemic corticosteroid treatments.

Many patients with neurosarcoidosis have already initiated treatment for sarcoidosis before they present with neurologic symptoms, which renders MRI less sensitive, as the classic imaging findings of neurosarcoidosis may be resolving or resolved. SWI provides a unique utility for the efficient diagnosis of NS [1,5,22]. This study demonstrates that SWI is a useful tool for early diagnosis of neurosarcoidosis and expedited treatment. More information is still needed to determine if deep vein engorgement findings on SWI are indicative of prognostic factors as well as specificity for neurosarcoidosis.

In the future, SWI may be useful in monitoring patients' disease progression pre- and post-treatment. More research will need to be conducted to better demonstrate the

sensitivity and specificity of SWI in diagnosing neurosarcoidosis, but this case series presents supporting evidence of the utility of the medullary vein sign on SWI sequence. The results of this study should prompt health care professionals to consider using this sign to aid in the diagnosis of neurosarcoidosis in the appropriate clinical setting.

CSF analysis in all confirmed cases of neurosarcoidosis was largely an unimportant part of the diagnosis. CSF ACE was elevated in three out of ten confirmed neurosarcoidosis cases.

Limitations to this study are mostly due to the small number of patients we were able to find with confirmed neurosarcoidosis. The sensitivity and specificity calculated in this study are thus likely not a true measure of the diagnostic potential of the medullary vein sign in SWI sequence for neurosarcoidosis. It is likely that the sensitivity of the medullary vein sign would remain low; however, the specificity of this imaging finding is likely even higher. Other limitations to the study are due to the retrospective nature of the study. The neuroradiologist reviewed images back in time to assess for a specific sign in patients who were already known to have neurosarcoidosis, causing potential recall bias. Though this is a drawback to the study, the lack of literature on this topic prevents a prospective study from being performed in an efficient manner. More research is necessary to confirm the diagnostic value of the medullary vein sign. This study supports the use of the medullary vein sign on SWI imaging sequences as part of the criteria in diagnosing neurosarcoidosis in patients with and without other organ involvement of sarcoidosis.

**Supplementary Materials:** The following supporting information can be downloaded at: https://www.mdpi.com/article/10.3390/neurolint14030052/s1, Figure S1. MRI axial T1 images 1(a) of the gadolinium-enhanced brain demonstrating dural thickening and enhancement of the right tentorium and cerebellopontine angle (blue and red arrow).

**Author Contributions:** Conceptualization: S.S. Drafting the manuscript: R.L., E.K., A.B.M., R.R., P.F., J.J., J.W. and S.S.; Editing and Final Draft: S.S. All authors have read and agreed to the published version of the manuscript.

**Funding:** This research received no external funding.

**Institutional Review Board Statement:** Informed consent waived as this study was conducted under approval of West Virginia University IRB; IRB protocol number: 2008098307.

**Informed Consent Statement:** Patient consent was waived as this work contains no identifiable information or images of included patients used and work was IRB approval from West Virginia University obtained.

**Data Availability Statement:** Not applicable.

**Acknowledgments:** West Virginia Clinical and Translational Science Institute, Morgantown, WV; SS supported in part by WVCTSI via US National Institute of General Medical Sciences of National Institute of Health under award under 5U54GM104942-05.

**Conflicts of Interest:** The authors declare that the research was conducted in the absence of any commercial or financial relationships that could be construed as a potential conflict of interest.

## List of Abbreviation

| | |
|---|---|
| MRI | Magnetic Resonance Imaging |
| CSF | Cerebrospinal fluid |
| EVD | External Ventricular drain |
| ACE | Angiotensin converting enzymes |
| LP | Lumbar Puncture |
| TTE | Transthoracic Echocardiography |
| ICP | Intracranial Pressure |
| CNS | Central Nervous System |
| NMO | Neuromyelitis Optica |
| MS | Multiple Sclerosis |

|       |                                            |
|-------|--------------------------------------------|
| CN    | Cranial Nerve                              |
| SWI   | Susceptibility weighted imaging            |
| MRI   | Magnetic resonance imaging)                |
| OSA   | obstructive Sleep apnea                    |
| GERD  | Gastroesophageal reflux disease            |
| IVDU  | Intravenous drug use                       |
| SE    | spin-echo                                  |
| MP-RAGE | magnetization prepared rapid gradient echo |
| MIP   | maximum intensity projection               |
| MinIP | minimum intensity projection               |

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
