# Peer review of "Relevance of Medullary Vein Sign in Neurosarcoidosis"

_2035-8377, doi:10.3390/neurolint14030052_

Round 1

Reviewer 1 Report

The authors of this article describe the findings of medullary vein signs on MRI brain scans in a retrospective cohort of suspected neurosarcoidosis (NS) patients. The NS patients were distributed as 7 definitive, 5 probable, and 16 possible NS. Furthermore, four patients in had infectious meningoencephalitis. Medullary vein sign was found in seven patients.

The limitation of this study is the retrospective design and that only 12 patients had biopsy verification of granulomatous inflammation. However, as NS is rare and difficult to diagnose, I find the article very interesting.

Specific comments to address are:

Introduction

Line 43: “Neurosarcoidosis affects 3-10%”. However, it is confusing that the discussion section says 30% (line 150).

Line 64-65: Likewise, it is confusing that the authors wrote ”high levels of ACE.. is common” and in line 66 wrote, ”only 33-50%.... elevated CSF ACE..”.  Stern et al. 2018 (your ref. 6) do not recommend using it.

Line 70:  New guidelines are published by Baughman et al. 2021 (ERS clinical practice guidelines on treatment of sarcoidosis).

Line 86: Consider adding your aim to the end of the introduction.

Method and 3. Result.

Line 91 and 99-100: information about cervical biopsy. Could it be moved to section 3.1, where the rest of the biopsy information is?

Please define or rephrase “confirmed” In line 90 and 92.

General Results

Line 109: Was there only information about sex and symptoms in 24 patients? Why do you write 32 possible/definitive neurosarcoidosis cases when you end up with 28 NS patients and 4 patients with another diagnosis?

Line 111: Is it possible to add frequency percentages to some symptoms? It gives a better impression of the cohort. It is also surprising that 30% did not receive any immunosuppression.

Line 119 and Table 1:  

In Table 1:   please define confirmed NS in the headline. Do you mean definite NS? Otherwise, it would really be helpful, If you add a column with their NS diagnosis: definite NS, probably NS, possible NS.

In Table 1:  Patient number 5: you wrote Imuran.

Where are the last two patients with medullary vein sign, and their category probably NS, possible NS, or other diagnoses?

Did any of the patients have micro bleeding alone with medullary vein signs?

Line 119: I am sorry, but I do not understand your calculation of sensitivity and specificity. Clarify it, please, but is it necessary? If I understand table 1 correctly, then 5 (71%) out of 7 definite NS patients have medullary vein signs, and it is a strong result.

MRI findings

Line 135: Likewise, some comments about the frequency of MRI findings, e.g., normal scans, involvement of the brain, spinal cord, etc. It would give a better impression of the cohort.

Minor comments:

Line 11: spelling error in “depratment”

Line 263: reference article number missing in “Ungprasert P, Matteson EL …”

Author Response

Manuscript ID: neurolint-1860070

Neurology International

Title: Relevance of Medullary Vein Sign in Neurosarcoidosis

We are grateful towards the Editor for considering our manuscript and would like to thank the Reviewers for the time they invested in improving the quality of our paper through their valuable suggestions. Please see attached point-by-point corrections or explanations to the Reviewers' comments. 

The authors of this article describe the findings of medullary vein signs on MRI brain scans in a retrospective cohort of suspected neurosarcoidosis (NS) patients. The NS patients were distributed as 7 definitive, 5 probable, and 16 possible NS. Furthermore, four patients in had infectious meningoencephalitis. Medullary vein sign was found in seven patients.

The limitation of this study is the retrospective design and that only 12 patients had biopsy verification of granulomatous inflammation. However, as NS is rare and difficult to diagnose, I find the article very interesting.

 Thank you for your kind words and positive feedback. We highly appreciate the Reviewers guidance and thank them for their time.

Specific comments to address are:

Introduction

R#1.1: Line 43: “Neurosarcoidosis affects 3-10%”. However, it is confusing that the discussion section says 30% (line 150).

A#1.1: Thank you for this suggestion. We now have made the changes in manuscript as suggested. Please refer to the manuscript.  

R#1.2: Line 64-65: Likewise, it is confusing that the authors wrote ”high levels of ACE.. is common” and in line 66 wrote, ”only 33-50%.... elevated CSF ACE..”.  Stern et al. 2018 (your ref. 6) do not recommend using it.

A#1.2: Thank you for this suggestion. We now have made the changes in manuscript as suggested. Please refer to the manuscript.  

Also, we have replaced stern et.al reference with new reference

R#1.3: Line 70:  New guidelines are published by Baughman et al. 2021 (ERS clinical practice guidelines on treatment of sarcoidosis).

A#1.3: Thank you for this suggestion. We now have made the changes in manuscript as suggested. Please refer to the manuscript.  

R#1.4: Line 86: Consider adding your aim to the end of the introduction.

 A#1.4: Thank you for this suggestion. We now have made the changes in manuscript as suggested. Please refer to the manuscript.  

Method and 3. Result.

R#1.5: Line 91 and 99-100: information about cervical biopsy. Could it be moved to section 3.1, where the rest of the biopsy information is?

A#1.5: Thank you for this suggestion. We now have made the changes in manuscript as suggested. Please refer to the manuscript.  

R#1.6: Please define or rephrase “confirmed” In line 90 and 92.

A#1.6: Thank you for this suggestion. We now have made the changes in manuscript as suggested. Please refer to the manuscript.  

General Results

R#1.7: Line 109: Was there only information about sex and symptoms in 24 patients? Why do you write 32 possible/definitive neurosarcoidosis cases when you end up with 28 NS patients and 4 patients with another diagnosis?

A#1.7: Thank you for this suggestion. We now have made the changes in manuscript as suggested. Please refer to the manuscript.  

R#1.8: Line 111: Is it possible to add frequency percentages to some symptoms? It gives a better impression of the cohort. It is also surprising that 30% did not receive any immunosuppression.

A#1.8: Sure, now we have added percentage value for the symptoms. Also, for the treatment part it was not available for all the cases as patients were moved to other facility after initial diagnosis and had follow up at outside facility. Please refer to the manuscript.  

Line 119 and Table 1:  

Reviewer 1:

R#1.9: In Table 1:   please define confirmed NS in the headline. Do you mean definite NS? Otherwise, it would really be helpful, If you add a column with their NS diagnosis: definite NS, probably NS, possible NS.

A#1.9: Yes, we now have indicated this as * definitive neurosarcoidosis cases and ** were for two cases of possible neurosarcoidosis. Please see Table 1

R#1.10: In Table 1:  Patient number 5: you wrote Imuran.

A#1.10: Thank you pointing this out its azathioprine we made the correction. Please refer to the manuscript.  

R#1.11: Where are the last two patients with medullary vein sign, and their category probably NS, possible NS, or other diagnoses?

A#1.11: The rest of 2 cases in table 1 are possible neurosarcoidosis cases. Please refer to the manuscript.  

R#1.12: Did any of the patients have micro bleeding alone with medullary vein signs?

A#1.12: No none of the cases had any signal drop out on GRE and there was no microbleed.

R#1.13: Line 119: I am sorry, but I do not understand your calculation of sensitivity and specificity. Clarify it, please, but is it necessary? If I understand table 1 correctly, then 5 (71%) out of 7 definite NS patients have medullary vein signs, and it is a strong result.

A#1.13: The sensitivity of the medullary vein sign in this study was 71.42% and the specificity was 92.3%. We calculated Sensitivity was calculated as the number of correctly identified cases of neurosarcoidosis based on medullary vein sign divided by the total number of patients in this study that were found to have confirmed neurosarcoidosis. Specificity was calculated as the number of patients in the study who did not have neurosarcoidosis or the medullary vein sign divided by all individuals without neurosarcoidosis.

MRI findings

R#1.14: Line 135: Likewise, some comments about the frequency of MRI findings, e.g., normal scans, involvement of the brain, spinal cord, etc. It would give a better impression of the cohort.

  A#1.14: Thank you for your comments because of the limitation of the numbers of the images as the journal guidelines we did not added other scan earlier. But now we have these as Figure S1.

Minor comments:

R#1.15: Line 11: spelling error in “depratment”

 A#1.15: Thank you for pointing this we now have made the correction.

R#1.16: Line 263: reference article number missing in “Ungprasert P, Matteson EL …”

A#1.16: Thank you for pointing this we now have made the correction.

Reviewer 2 Report

Fig. 1 is completely unconvincing. Expanded image with focus to the dilated medullary vein is necessary. But, its finding is unconvincing for readers. Radiological significance of this finding for the diagnosis of nerosarcoidosis is not understandable.

 In discussion, description of the venous malformation seems to be necessary.

Author Response

Manuscript ID: neurolint-1860070

Neurology International

Title: Relevance of Medullary Vein Sign in Neurosarcoidosis

We are grateful towards the Editor for considering our manuscript and would like to thank the Reviewers for the time they invested in improving the quality of our paper through their valuable suggestions. Please see attached point-by-point corrections or explanations to the Reviewers' comments. 

The authors of this article describe the findings of medullary vein signs on MRI brain scans in a retrospective cohort of suspected neurosarcoidosis (NS) patients. The NS patients were distributed as 7 definitive, 5 probable, and 16 possible NS. Furthermore, four patients in had infectious meningoencephalitis. Medullary vein sign was found in seven patients.

The limitation of this study is the retrospective design and that only 12 patients had biopsy verification of granulomatous inflammation. However, as NS is rare and difficult to diagnose, I find the article very interesting.

 Thank you for your kind words and positive feedback. We highly appreciate the Reviewers guidance and thank them for their time.

Reviewer 2:

Comments and Suggestions for Authors

R#2.1: Fig. 1 is completely unconvincing. Expanded image with focus to the dilated medullary vein is necessary. But, its finding is unconvincing for readers. Radiological significance of this finding for the diagnosis of nerosarcoidosis is not understandable.

A#2.1: Thank you for your comments now we have updated the images and added them as Figure 1. Please refer to the manuscript

 R#2.2: In discussion, description of the venous malformation seems to be necessary.

A#2.2: Thank you now we have added this in our discussion. Please refer to the manuscript.

Reviewer 3 Report

This retrospective study tested the sensitivity and specificity of the “medullary vein sign” in 32 patients with possible sarcoidosis. The authors find a 60% sensitivity and 92% specificity. I find there are a few methodological details missing, and that the overall organisation could be improved, as I described below.

More detail on the MRI acquisition would be useful, such as the scanner field strength and manufacturer, sources of the imaging data, scan parameters. Were the SWI parameters consistent across the study? Also add information for the other sequences that are reported in the results but not the methods e.g. MPRAGE, PC-T1.

 “In this study, MRI images were reviewed using SWI/GRE sequences in a retrospective search for medullary vein engorgement, termed medullary vein sign…” This paragraph should be in methods. Can the authors give any more details on the image review e.g. who retrospectively reviewed the images, single rater or multiple, was any quantitative processing applied?

Since tests included in sarcoidosis diagnosis lack specificity and sensitivity, and since sarcoidosis is clinically and aetiologically heterogeneous, a greater focus should be given to understanding why that is the case – I suggest that sarcoidosis might really be a family of diseases and not a single entity, and that unsupervised learning approaches (PCA, clustering) could be used to empirically identify new subgroups that are individually easier to diagnose with a specific set of tests.

Minor:

·        Please describe the statistical approach and tests used, not just the variables included.

·        In the discussion can the authors give attention to the cases that had the medullary vein sign but did not have neurosarcoidosis?

·        Please report the sensitivity and specificity of medullary vein sign in the abstract.

·        Table 1: be consistent in the presentation, e.g. “lymphocytes” vs “lymphs”, “Monocytes” vs “Monos”.

·        Abstract: “Seven patients among all of 27 these cases were found to have the medullary vein sign on imaging” – please clarify which patient groups these 7 belonged to.

·        Introduction: Can the authors be a little more specific than “Sarcoidosis is a relatively rare disease that affects greater than 150,000 Americans”? What’s the upper bound of the estimate?

Author Response

Manuscript ID: neurolint-1860070

Neurology International

Title: Relevance of Medullary Vein Sign in Neurosarcoidosis

We are grateful towards the Editor for considering our manuscript and would like to thank the Reviewers for the time they invested in improving the quality of our paper through their valuable suggestions. Please see attached point-by-point corrections or explanations to the Reviewers' comments. 

The authors of this article describe the findings of medullary vein signs on MRI brain scans in a retrospective cohort of suspected neurosarcoidosis (NS) patients. The NS patients were distributed as 7 definitive, 5 probable, and 16 possible NS. Furthermore, four patients in had infectious meningoencephalitis. Medullary vein sign was found in seven patients.

The limitation of this study is the retrospective design and that only 12 patients had biopsy verification of granulomatous inflammation. However, as NS is rare and difficult to diagnose, I find the article very interesting.

 Thank you for your kind words and positive feedback. We highly appreciate the Reviewers guidance and thank them for their time.

Reviewer 3:

Comments and Suggestions for Authors

This retrospective study tested the sensitivity and specificity of the “medullary vein sign” in 32 patients with possible sarcoidosis. The authors find a 60% sensitivity and 92% specificity. I find there are a few methodological details missing, and that the overall organisation could be improved, as I described below.

R#3.1: More detail on the MRI acquisition would be useful, such as the scanner field strength and manufacturer, sources of the imaging data, scan parameters. Were the SWI parameters consistent across the study? Also add information for the other sequences that are reported in the results but not the methods e.g. MPRAGE, PC-T1.

A#3.1: Thank you for your comments because of the limitation of the numbers of the images as the journal guidelines we did not added other images earlier. But now we have these as Figure S1.

The scanner used were all GE and simens 3 tesla machine. Again, these were data obtained from clinic cases and it was SWI parameters were not consistent across the cases.

R#3.2: “In this study, MRI images were reviewed using SWI/GRE sequences in a retrospective search for medullary vein engorgement, termed medullary vein sign…” This paragraph should be in methods. Can the authors give any more details on the image review e.g. who retrospectively reviewed the images, single rater or multiple, was any quantitative processing applied?

A#3.2: We have added this in method also, we have added limitation in discussion section that this was retrospective data obtained from clinical imaging data. Further study is needed to look in to significance and yield of medullary vein sign in the diagnostic neuroimaging biomarker for the neurosarcoidosis. The images were reviewed by two neuroradiologist and this was multiple rater. No quantitative processing was applied. This paper is in relevance of medullary vein sign a single center experience and not in overall specificity or sensitivity of the medullary vein sign for diagnostic purposes in the sarcoid and future study will be needed for this

R#3.3: Since tests included in sarcoidosis diagnosis lack specificity and sensitivity, and since sarcoidosis is clinically and aetiologically heterogeneous, a greater focus should be given to understanding why that is the case – I suggest that sarcoidosis might really be a family of diseases and not a single entity, and that unsupervised learning approaches (PCA, clustering) could be used to empirically identify new subgroups that are individually easier to diagnose with a specific set of tests.

A#3.3: This paper is meant to address the relevance of the imaging finding known as medullary vein sign as a useful tool in the diagnosis of potential neurosarcoidosis. The paper is not meant to ponder a single or multiple mechanism of disease contributing to the pathogenesis of neurosarcoidosis. Also, we have added limitation in discussion section that this was retrospective data obtained from clinical imaging data. Further study is needed to look in to significance and yield of medullary vein sign in the diagnostic neuroimaging biomarker for the neurosarcoidosis.

Minor:

  • Please describe the statistical approach and tests used, not just the variables included.

      Thank you for the comments now we have updated this in our manuscript.

  • In the discussion can the authors give attention to the cases that had the medullary vein sign but did not have neurosarcoidosis?

      Unfortunately, due to limitation of the length of manuscript based on journal guidelines we are not able to add all other cases here.

  • Please report the sensitivity and specificity of medullary vein sign in the abstract.

     Thank you for the comments now we have updated this in our abstract.

  • Table 1: be consistent in the presentation, e.g. “lymphocytes” vs “lymphs”, “Monocytes” vs “Monos”.

     Thank you for the comments now we have updated this in our Table.

  • Abstract: “Seven patients among all of 27 these cases were found to have the medullary vein sign on imaging” – please clarify which patient groups these 7 belonged to.

     Thank you for the comments now we have updated this in our manuscript.

  • Introduction: Can the authors be a little more specific than “Sarcoidosis is a relatively rare disease that affects greater than 150,000 Americans”? What’s the upper bound of the estimate?

The sentence here is just to introduce a general idea about the prevalence of neurosarcoidosis as compared to other neurological diseases. It would be trivial to give an exact upper limit for the purposes of this paper. The important aspect of this sentence is to give sense as to why clinical suspicion is a vital part of the diagnosis. 

Round 2

Reviewer 2 Report

Revision is almost good for publication.